# Dual-Scale Doppler Attention for Human Identification

**DOI:** 10.3390/s22176363

**Published:** 2022-08-24

**Authors:** Sunjae Yoon, Dahyun Kim, Ji Woo Hong, Junyeong Kim, Chang D. Yoo

**Affiliations:** 1School of Electrical Engineering, Korea Advanced Institute of Science and Technology, Daejeon 34141, Korea; 2Department of AI, Chung-Ang University, Seoul 06974, Korea

**Keywords:** deep learning, human identification, micro-Doppler radar, fine-grained feature analysis

## Abstract

This paper considers a Deep Convolutional Neural Network (DCNN) with an attention mechanism referred to as Dual-Scale Doppler Attention (DSDA) for human identification given a micro-Doppler (MD) signature induced as input. The MD signature includes unique gait characteristics by different sized body parts moving, as arms and legs move rapidly, while the torso moves slowly. Each person is identified based on his/her unique gait characteristic in the MD signature. DSDA provides attention at different time-frequency resolutions to cater to different MD components composed of both fast-varying and steady. Through this, DSDA can capture the unique gait characteristic of each person used for human identification. We demonstrate the validity of DSDA on a recently published benchmark dataset, IDRad. The empirical results show that the proposed DSDA outperforms previous methods, using a qualitative analysis interpretability on MD signatures.

## 1. Introduction

Human Identification (HI) serves as an essential building block for many personal identification services including surveillance, security and identification systems. In general, HI adopts visual video as it has information that can be easily understood. However, HI through visual video can be problematic under low lighting conditions and with the privacy infringement issues [1]. As an alternative to the use of cameras, radar devices bypass these problems. Radar is a detection system that measures the distance, direction, angle, and speed of a target by emitting electromagnetic waves from a device and analyzing electromagnetic waves that are reflected and returned from an object. Radar can operate under low light conditions, and its tendency to bend around obstacles makes it suitable for identification in obscured environments [2,3,4]. Moreover, radar information is relatively safe regarding privacy, as people cannot directly interpret information obtained by radar. Furthermore, radar also has the advantage of observing far-distance targets. Overall, radar is a more robust sensor for HI than visual video. However, since it is difficult to observe the iris, voice, and face using radar, this paper adopts gait instead of aforementioned biometrics. Gait can be observed from a distance, and unlike the biometrics (iris, voice, and face), it is behavior over a certain period of time, so the security is relatively high. These advantages are consistent with the reason for using radar-based HI instead of video-based HI.

For this radar-based HI, the micro-Doppler (MD) signature has been a popular choice. As shown in Figure 1, it sequentially records Doppler effects in electromagnetic signals of moving targets [5], and the superposition of these Doppler signals is summarized to make MD signatures, where it holds granularity to specify their information. In the methodology, conventional machine learning (ML) algorithms have attempted to analyze the statistics of the MD signatures [6,7,8,9]. Unfortunately, ML has several drawbacks as it is based on heuristic feature extraction and a low capacity model. Recent Deep Convolutional Neural Network (DCNN), via capturing spatial relationships and comprehending high-level spatial features, overcomes these limitations and has revolutionized many applications including radar-based human identification and motion recognition. Motion recognition is easily conducted by training DCNN to recognize the common patterns among identical motions. However, human identification [10,11] should identify the unique characteristics of each person in an uncontrolled scenario, which requires more fine-grained understanding than motion recognition.

For these fine-grained understandings, we exploit that MD signatures from different body parts such as arms, legs, and torso display different signal characteristics. Each of us has a unique gait characteristic distinguished by different sized body parts moving and swinging in a distinctive pattern, as the arms and legs move rapidly, while the torso moves slowly. Our empirical analysis validates that more than 95% of MD signatures on the radar dataset (Validation is performed on the IDRad dataset) include this distinguishable gait characteristic. Therefore, we utilize these unique gait characteristics from the MD signatures to identify humans. For the detailed explanation of the gait characteristics of MD signatures, as shown in Figure 1, MD signatures have fast and slow-moving components. The MD signatures recorded the Doppler effects along the time, where the orange curve shows a small amplitude wave denoting slow-moving features, and the signal that spreads around the orange curve makes a high-amplitude wave representing fast-moving features.

Under this observation, our proposed Dual-Scale Doppler Attention (DSDA) is composed of Temporal Window Attention (TWA), which attends fast-moving MD signatures via building temporal windows for MD signatures, and Holistic Window Attention (HWA), which attends slow-moving MD signatures via building holistic windows. To perform TWA and HWA, we define a common attention module defined as Window Attention (WA), which conducts self-attention among windows by calculating their similarities. In the overall pipeline, TWA extracts fast-moving features by generating multiple temporal windows and applying WA among them. HWA generates slow-moving features by subtracting the attended fast-moving features from original signals, and WA again performs self-attention between subtracted and original signals. With these attended fast and slow-moving features, DSDA extracts unique characteristics of each person in fast and slow moving, and identifies human identity. Using the IDRad [10] dataset, we validate state-of-the-art performances on human identification tasks and show the interpretability of the proposed DSDA.

## 2. Related Works

### 2.1. Doppler Radar Systems for Human Identification

Doppler radar, which uses a single-tone radio wave, has been frequently used for human identification [6,10,12,13]. By the Doppler effect, the received frequency $f_{r}$ of the moving target is shifted away from the transmitted frequency $f_{t}$, and the Doppler frequency is defined by subtracting $f_{t}$ from $f_{r}$ as:(1)$$\begin{array}{c} f_{r}=f_{t}\left(1+v/c\right)/\left(1-v/c\right), \end{array}$$(2)$$\begin{array}{c} f_{d}=f_{r}-f_{t}=2vf_{t}/\left(c-v\right), \end{array}$$
where *c* is the speed of light, and *v* is the radial speed of the moving target. By capturing the Doppler shift, it is able to detect the human motions [5,14,15]. Moreover, frequency-modulated continuous-wave (FMWC) radar is commonly used for short-range multiple targets detection by generating a Doppler map within a certain range [10,16,17]. Vandersmissen et al. [10] utilized low-power 77 GHz FMCW radar for person identification and constructed the IDentification with Radar (IDRad) data set. IDRad is a micro-Doppler map received from several people walking around spontaneously in any possible direction. Our proposed DSDA is experimented on IDRad and validates its interpretability on MD signatures.

### 2.2. Deep Learning for Micro-Doppler Signatures

As the use of radar-based systems increases, several applications of MD signatures using deep learning have emerged [18,19,20,21]. Kim et al. [22] first applied a neural network for human motion recognition on MD signatures and showed the applicability of deep learning on radar signals. After that, several deep learning techniques have been applied for the radar-based motion recognition, including large-scale pre-training [23] and recurrent neural network [24]. Lin et al. [25] proposed iterative CNN followed with random forests in MD signatures which showed performance boost. Park et al. [23] utilized DCNN pre-trained with a large-scale image classification dataset, ImageNet [26], which presented the connectivity between radar and computer vision. Furthermore, Wang et al. [24] used a recurrent neural network (RNN) to detect dynamic gestures with a short-range radar-based sensor, Google’s Soli. Recent studies have been conducted via performing human identification (HI) on an MD signature, which requires the understanding of unique characteristics in a single person. In detail, MD signatures from heartbeat signals are utilized for HI [11,27]. Henceforth, MD signatures on gait characteristics of humans [10] are used for this HI, which is more challenging as they are performed in an uncontrolled scenario where a target is allowed to walk around in a free and spontaneous way. Cao et al. [27] primarily applied DCNN to MD signatures for HI. Vandersmissen et al. [10] also used the DCNN and constructed public dataset IDRad for HI, which contributed to subsequent research. Although several methods have been proposed for the aforementioned tasks, they do not fully utilize the details of MD signatures induced from moving human body parts. Therefore, we propose DSDA, which can exclusively recognize unique signals generated by human walking.

### 2.3. Recent Radar-Based Human Identification Analysis

Radar systems have mainly been applied on the radar-based human identificati-ons [10,11,27,28,29]. We also compare these previous works to our research in terms of the differences, advantages and disadvantages. The work [28] is performed on detecting humans in specific conditions (i.e., short-range through-wall and long-range foliage penetration). The difference between this study and ours is that it was carried out to find people under specific conditions, but our proposed DSDAs are more contributing on human identification from general human behaviors including human walking motions, arm and body movements. In terms of the method, they applied SVM for human detection. However, our study also contain SVM methods and makes more experimental contributions (SVM, CNN, RNN, and Attention model). The advantage of this study, we think, is that it defines specific tasks well and suggests their solutions. However, the disadvantage is that it is too task-specific and reduces its applicability for other research.

The work [29] holds the commonality with our work in that it performs human identification through the recognition of gait characteristics in MD signatures. However, this work mainly focused on open-set feature analysis, which means how the model can do better when the ‘unknown class’ exists in the inference, but our research is mainly about sequential feature pattern analysis; thus, our experimental contributions are more relying on sequential pattern analysis and its solutions. The advantage of the work [29] is novel problem definition for establishing the generality of human identification, but the disadvantage is that the feature analysis is insufficient in that the MD signatures contain sequential information.

The works in [11,27] are also holding commonality with our work in that they perform feature pattern analysis (limbs, torso, heart beat). However, the difference is that our model introduces an attention method to better recognize sequential feature patterns. As shown in Table 1, in the paper, our initial attempts also include the CNN models such as [10,11,27], but we confirmed that the RNN-based model should perform better. Therefore, our further studies are focused on designing sequential feature processing model (RNN and DSDA). The advantages of works [11,27] are specific feature pattern analysis, and we speculate that the disadvantages are a lack of concern for a model that can recognize the feature pattern well because of reliance on the popular CNN model.

For the work [10], our proposed DSDA is validated on the same dataset (IDRad) released in [10]. However, the simple convolutional neural network used in [10] is limited in understanding sequential patterns in the radar continuity; thus, we more focused on a method that can accurately recognize sequential information of the radar sequence. In this respect, we have performed several experimental contributions including Recurrent Neural Network models and an Attention model. Finally, we proposed a Dual-Scale Doppler Attention technique for recognizing fast and slow-moving patterns that are prominently present in continuous signals, where we speculate that this makes the methodological differences comparing to the work [10]. The advantages in the work [10] exist in the contributions from dataset release and task proposal, but for the disadvantages, we guess that there is a lack of contribution on how to better understand the features pattern in sequential MD signatures.

## 3. Method

### 3.1. Generating Micro-Doppler Signatures

Our proposed Dual-Scale Doppler Attention (DSDA) is validated on the Micro-Doppler signatures (MD signatures); we first explain how to generate the MD signatures. To make 45 time stamps of MD signatures, we first build a single time stamp MD signature and connect 45 stamps in a row. As shown in Figure 2, the single time stamp MD signature is obtained from a single range-Doppler map. We integrate the range-Doppler map (e.g., the dimension for the range-Doppler map is 256 × R, where 256 is the Doppler channel and R is the number of discrete range bins) along the range axis, which constructs the single MD signature (e.g., 256 × 1).

This single MD signature contains the Doppler-shift information, where this Doppler-shift value denotes whether the target is moving closer or farther away (e.g., 129–256 channels contain the Doppler shift by the target moving closer and 1–128 channels contain the Doppler shift by the target moving farther away). Following the [10], some channels (i.e., 127–129 channels are zero-Doppler effective, and Doppler channels at both ends are too noisy) are not helpful to identify the human. We also remove them and finally can generate 205 × 45 time stamps MD signatures.

### 3.2. Model Overview

MD signatures are provided for identifying the human identity [10,11,27,28,29,30,31]. DSDA takes MD signature $R\in R^{C\times T}$ consisting of the number of channels *C* and the number of time stamps *T* (C=205, T=45). The time stamp can be modulated according to the size of MD signatures. As shown in Figure 3, the MD signature is a time-evolving sequence representing the micro-Doppler effect of a person moving.

Figure 3 gives a schematic of DSDA consisting of Window Generation, Temporal Window Attention (TWA) and Holistic Window Attention (HWA). Window Generation provides dual-scale windows: a temporal window and holistic window. The temporal window $W_{T}$ is focused on recognizing fast-moving features extracted by MD signatures so that it is uniformly divided with a temporal sliding window of stride *S*. The temporal window size is C×L; as such, the total number of temporal windows is N (S=5, L=25 and N=5). The holistic window $W_{H}$ is focused on recognizing the slow-moving signal and extracted by an original MD signature. For the Window Encoder (WE), it embeds temporal windows and holistic window into *d*-dimensional feature representation through a series of Conv-ELU-MaxPooling layers. The encoded temporal features T and holistic features H are defined as:(3)$$\begin{array}{cc} T & =\mathrm{LN}\left(\mathrm{MaxPool}\left(\mathrm{ELU}\left(\mathrm{Conv}\left(W_{T}\right)\right)\right)\right)\in R^{N\times d}, \end{array}$$(4)$$\begin{array}{cc} H & =\mathrm{LN}\left(\mathrm{MaxPool}\left(\mathrm{ELU}\left(\mathrm{Conv}\left(W_{H}\right)\right)\right)\right)\in R^{1\times d}, \end{array}$$
where LN denotes the layer normalization [32] and the Exponential Linear Unit (ELU) [33] is nonlinearity operation. We present detailed specifications of the Window Encoder used in TWA and HWA in Table 1 and Table 2. The number of temporal windows is considered as N = 3 in Table 1 (i.e., this is designed to help understand the Window Encoder’s structure used in the Temporal Window Attention). The following sub-sections will explain the details of the remaining model components.

### 3.3. Window Attention

Our proposed basic attention unit of the TWA and HWA is referred to as Window Attention (WA), which calculates the attention weights among multi-element features. Given $X={\left[x_{1}\ldots x_{I}\right]}^{T}\in R^{I\times d}$ and $Y={\left[y_{1}\ldots y_{J}\right]}^{T}\in R^{J\times d}$, the WA $W\left(X,Y\right):R^{I\times d}\times R^{J\times d}\to R^{I\times d}$ attends *X* using *Y*, which is defined as follows:(5)$$\begin{array}{cc} & w\left(x_{i},Y\right)=\sum_{j=1}^{J}\left(\mathrm{ELU}\left(W_{1}x_{i}\right)\mathrm{ELU}{\left(W_{2}y_{j}\right)}^{T}\right), \end{array}$$(6)$$\begin{array}{cc} & W\left(X,Y\right)=\mathrm{Softmax}\left(\left[w\left(x_{1},Y\right)\;\cdots \;w\left(x_{I},Y\right)\right]\right)X, \end{array}$$
where $w\left(x_{i},Y\right):R^{d}\times R^{J\times d}\to R$ and $W_{1},W_{2}\in R^{d\times d}$ are learnable parameters. (Our experiments also validate the attention method in the Section 4.4) Using the ELU nonlinearity, WA tries to preserve common signatures between input *X* and *Y* windows. Based on the WA, the following TWA and HWA are defined and attend their moving signatures. We also provide the process of Window Attention in Figure 4, where WA is performed on two window features. For the TWA, the two types of window are temporal windows and, for the HWA, the two types are holistic windows.

### 3.4. Temporal Window Attention

We observe that the fast-moving features of arms and legs are unique depending on the person’s height and walking pattern. To preserve and highlight these features, the Temporal Window Attention (TWA) takes multiple temporal windows as inputs and applies WA as follows:(7)$$\begin{array}{c} T\leftarrow \mathrm{LN}(W(T,T)+T), \end{array}$$(8)$$\begin{array}{c} T_{avg}=\mathrm{AvgPool}\left(T\right)\in R^{1\times d}. \end{array}$$
where AvgPool() is the average pooling function. TWA produces two types of features. One is the original TWA output T in Equation (Equation 7), which contributes to classifying the person identification, and the other is the average of T over *N* temporal windows $T_{avg}$ in Equation (Equation 8). This average of temporal window attended features $T_{avg}$ is treated to have the characteristic of the overall fast-moving features. Furthermore, $T_{avg}$ is used to remove fast-moving information from the original holistic features in the following Holistic Window Attention.

### 3.5. Holistic Window Attention

The Holistic Window Attention (HWA) is designed to recognize slow-moving signatures such as the torso. Different from TWA, HWA takes two types of features as input. One is original holistic features H, and the other is subtracted holistic features $H_{sub}$ obtained by subtracting the $T_{avg}$ from H in Equation (Equation 9). Semantically, the H can include slow and fast-moving features, and the $H_{sub}$ includes only the slow-moving features by removing characteristics of overall fast-moving features. These two holistic features are concatenated, and HWA attends the common slow-moving signals using the WA in Equations (10) and (11):(9)$$\begin{array}{cc} & H_{sub}=H-T_{avg}, \end{array}$$(10)$$\begin{array}{cc} & H^{*}=\left[HH_{sub}\right]\in R^{2\times d}, \end{array}$$(11)$$\begin{array}{cc} & H^{*}\leftarrow \mathrm{LN}\left(W\left(H^{*},H^{*}\right)+H^{*}\right). \end{array}$$
where [··] is the concatenation operation. Holistic features $H^{*}$ through HWA represent the slow-moving feature in the MD signature and are utilized to classify the targets.

### 3.6. Classification

In classification, we use two aforementioned temporal and holistic features for classifying targets. These two attended temporal and holistic features are concatenated and transmitted to the final classifier as follows:(12)$$\begin{array}{cc} & P=\left[TH^{*}\right]\in R^{(N+2)\times d}, \end{array}$$(13)$$\begin{array}{cc} & P^{*}=\mathrm{Flat}\left(P\right)\in R^{\left((N+2)\times d\right)}, \end{array}$$(14)$$\begin{array}{cc} & C=W_{c}\left(\mathrm{Dropout}\left(\mathrm{ELU}\left(W_{p}P^{*}\right)\right)\right)\in R^{C_{T}}, \end{array}$$(15)$$\begin{array}{cc} & p=(\mathrm{Softmax}(C)) \end{array}$$
where Flat() is a function that converts 2D data into 1D in order to apply the CNN data type to the fully connected neural network, Equation (Equation 14) is the last Dropout–ELU–Linear block in the classifier, $W_{p}\in R^{128\times ((N+2)\times d)}$ and $W_{c}\in R^{C_{T}\times 128}$ are the learnable parameters, $C_{T}=5$ (IDRad dataset contain five different targets made by five different people.) is the number of targets and p={p[1],p[2],p[3],p[4],p[5]} is inferred prediction distribution values. In the inference, prediction is performed using argmax function on the *p*. We also add some specifications of the classifier network in Table 3. The (3 + 2) in Table 3 denotes concatenation between TWA and HWA, where HWA includes (2 × 1024) features from the holistic feature (1 × 1024) in Table 2 and the subtracted feature (1 × 1024). TWA includes (3 × 1024) features in Table 3.

### 3.7. Training Loss

The entire model is trained in an end-to-end manner using cross-entropy loss $L_{HI}\left(y,p\right)$, where the *y* is the ground-truth target information and the *p* is the prediction of human identification from DSDA, as shown below:(16)$$\begin{array}{c} L_{HI}\left(y,p\right)=-log\left(p\left[y\right]\right) \end{array}$$

## 4. Experiments

### 4.1. IDRad Dataset

As the MD signature dataset for HI, we use IDRad [10] using FMCW radar, which records range-Doppler maps with a speed of around 15 FPS. The IDRad dataset contains 95,650 frames of 20 min for a training set and 22,535 frames of 5 min for a test set. One frame contains a Doppler frequency channel along the range axis. To construct micro-Doppler signatures, the IDRad dataset integrates a range-Doppler map along the range axis and connects the integrated Doppler signals in temporal order to make Doppler-time maps as previously stated [10]. Here, one time stamp includes 256 Doppler channels, and one MD signature is composed of 45 time stamps. For a fair comparison, we also performed the same preprocessing as [10] and used the default input of 205 × 45 MD signatures, which translates to 205 Doppler channels and 45 time stamps. Regarding the details of preprocessing, among 256 Doppler channels, 127∼129 Doppler channels representing static objects are removed and 24 Doppler channels in the top and bottom of the Doppler axis in MD signatures are also removed, because they are a too high speed range for humans to demonstrate. The IDRad dataset films the movements of five subjects whose age ranges are from 23 to 32 years old, weight ranges are from 60 to 90 kg and their height ranges are from 178 to 185 cm. Five subjects are able to move freely within a certain range of filmed room, and they are able to freely walk, run, or stop in various directions.

### 4.2. Experimental Details

The dimension of the hidden layer is set to d=1024. For DSDA, we use three blocks of the Conv-ELU-MaxPooling layer in the Window Encoder and four blocks of the Dropout [34]-ELU-Linear layer in the classifier. Our model can be easily implemented with six layers of convolutional neural networks and trained on NVIDIA TITAN V (12 GB of memory) GPU with an Adam optimizer [35] with $\beta_{1}=0.9,\beta_{2}=0.98$ and $\epsilon =10^{-9}$. For all experiments, we select the batch size of 64, a dropout rate of 0.2, and the model is trained up to 15 epochs. From this condition, we do not perform any hyperparameter fine tunning.

The overall evaluation of DSDA is performed using error rate, where the error rate is calculated as: $‘\mathrm{error}{\mathrm{rate}}^{’}=100$ × (numberofincorrectpredictions)/(totalsamples). The total sample can be composed of a validation set and test set.

### 4.3. Experimental Results

Table 4 summarizes the experimental results on the IDRad Dataset where we compare DSDA with several recent methods. Considering the temporal properties, we use 10 s MD signatures as an input (150 time stamps), which is different from the input of 3 s MD signatures (45 time stamps) in [10]. For the fair comparison, we also measured the baseline with the same sizes of the input MD signatures, where the baseline with 10 s MD signatures is reproduced using their public code and reported as ‘Baseline (150 time stamp)’ in Table 4. The extended input size of the baseline also shows slight effectiveness, but it still remained in the variation of original performances reported in the paper [10]. To validate the deep learning-based model on this task, we first built a PCA model with SVM classifier, which shows the classical classifying performance with a machine learning algorithm. Comparing to the performances of the PCA model, the deep learning-based models (i.e., baseline, LSTM-based model, DSDA) are giving superior performances. Although there are sequential data in the IDRad dataset, there have not been sequential models to perform human identification. Thus, for these sequential radar image data, we also consider a sequential RNN (Recurrent Neural Network) model on this task. We devised a Long Short-Term Memory (LSTM) [36,37] based model. The LSTM model shows better performances compared to the baseline built with a CNN structure, which explains the necessity of sequential understanding for radar data. The recent success of Transformer [38,39,40], our proposed DSDA, is based on the attention mechanism and also utilizes the specifications in radar (fast-moving and slow-moving features). DSDA achieves state-of-the-art performance against all methods with a large margin. The results indicate that extracting slow and fast-moving features and attending them improve the interpretability of MD signatures.

### 4.4. Ablation Study

We experiment with several variants of DSDA to measure the effectiveness of the proposed key components. The first block of Table 5 is full DSDA with three multi-scale windows composed of strides s={5,10,20} and windows L={35,25,25}. The second block of Table 5 provides ablation results of TWA and HWA. Since TWA and HWA have influenced performance improvements, both fast-moving features and slow-moving features imply target features. Especially as TWA boosts performance significantly, we can see that unique information, which identifies a person, is inherent in fast-moving features. The third block of Table 5 provides ablation results on the scale of temporal windows in TWA. Window size and stride between windows influence the extraction of the proper characteristic features. We have confirmed that the best performance is achieved when the stride *s* is 5 and window size *L* is 35 for single fixed window. The fourth block of Table 5 provides ablation results with multi-scale windows. Here, we use multi-scale temporal windows of various sizes for TWA via selecting several windows in the third block of Table 5. The *M* is the number of different types of windows. When M=3, we select the several combinations of three windows, where the best performances are shown with the windows composed of strides s={5,10,20} and windows L={35,25,25}. We also validate for M=4 via adding one more window on top of the best-performance condition in M=3. If all the results are not improved, then M=3 is the best-performance condition, so we report the averaging results on M=4. To adopt multi-scale temporal windows, TWA is applied to each window scale, and HWA uses several subtracted holistic features obtained by multi-scale temporal windows. We consider that several moving features extracted by multi-scale temporal windows make it suitable to capture a person’s unique characteristics.

Table 6 presents the ablation on the attention methods for $\mathrm{ELU}\left(W_{1}x_{i}\right)\mathrm{ELU}{\left(W_{2}y_{j}\right)}^{T}$ in Equation (Equation 5). We experimented with three attention methods that highlight common features. Based on the results, we selected A∗B for the final DSDA. Here, the operation; denotes concatenation with a *d*-dimensional axis. To follow the dimensional condition, we add more of an embedding matrix $W_{add}\in R^{d\times 1}$ for A+B and embedding matrix $W_{cat}\in R^{2d\times 1}$ for A;B. Our empirical experiments for more attention methods are in the variance of A+B and A;B, and they do not give further performance gain.

Table 7 represents the Kappa Index Analysis of our proposed DSDA with Baseline [10]. ‘TRUE’ means ‘correct’ on the target, and ‘FALSE’ means ‘incorrect’ on the target. To calculate the kappa value K=(1−Pa)/(Pa−Pc), where Pa is the observational probability of agreement and Pc is the hypothetical expected probability of agreement. Pa is obtained as Pa=(1082+122)/1490=0.808 and Pc is obtained as Pc=(1328/1490)×(1122/1490)+(162/1490)×(368/1490)=0.698. Thus, the kappa value is obtained as K=0.36. Therefore, according to the kappa value analysis, our proposed DSDA decision performance has ’fair’ strength of agreement with the baseline [10]. We consider this is because DSDA includes improved performance (10.8% error rate) compared to the baseline (24.7% error rate), which contributes to the disagreement with baseline in the case the baseline performs an incorrect decision on the target.

### 4.5. Qualitative Results

Figure 5 visualizes attention weights for temporal windows. For the input MD signature of 150 time stamps, attention weights are represented as a bar chart in the Figure 5. Here, temporal windows adopt 45 time stamps and 15 strides. Although the multi-scale windows performed well in this human identification task, we select the single window that can help easily understand how the attention weights of TWA are formed and these weights identify the fast-moving information of the MD signatures. Thus, we can find out which windows have been highlighted through the attention weight. The MD signatures corresponding to three windows (i.e., 1, 2, 3) from the left contain fast-moving features, and also, the signatures corresponding to two windows (i.e., 7, 8) from the right contain fast-moving features. Our analysis on synchronized video identifies that these fast-moving features are from the human walking. The other signatures (i.e., MD signatures from windows 4, 5, 6) are formed when the humans stop moving and change the direction of walking. The attention weights in TWA are highlighted on these windows containing fast-moving features such as walking; however, the small weights are given on slow-moving features such as changing direction or non-moving. Therefore, it is confirmed that the TWA is properly trained to recognize the fast-moving information from the MD signatures.

In Figure 6, we build a confusion matrix of DSDA. The vertical axis denotes the ground-truth category and the horizontal axis denotes the predicted category. For all targets, DSDA perform 100% accuracy on target 3 with the highest accuracy and performs 72% accuracy on target 4. We consider the reason why DSDA’s prediction is the lowest on target 4, where the target 4, target 1 and target 5 show similar movements and have a relatively similar body shape. This gives the challenge in identifying their characteristics. To qualitatively compare the confusion matrix with the baseline [10], we give the confusion matrix predicted from the baseline (We obtain confusion matrices from fully trained baseline and DSDA.) in the (b) of Figure 6. Comparing to the baseline, the performance improvement can be confirmed in all targets for our proposed DSDA, and it was confirmed that both the baseline and DSDA showed excellent performance in target 3. However, it is also notable that our DSDA is more improved in target 4, where this is because DSDA is more robust to distinguish fine-grained information in the MD signatures.

In Figure 7, we also perform a target-wise confusion matrix to calculate the sensitivity and specificity according to the targets. The average sensitivity is 0.88, and the average specificity is 0.97. Our DSDA is effective in the specificity, which means our model is highly sensible on the negative targets. It is also notable that DSDA performs 100% on target 3, which implies that the model finds suitable feature space that can classify target 3, and also, the model localizes other target features on the proper feature spaces.

In Figure 8, we perform efficiency analysis. The attention mechanism in DSDA does not require many memory resources (The resources that are required are joint space embedding matrices.) but also performs early saturation of training error rate. As shown in orange curve of Figure 8, the error rate on the IDRad training dataset converges more early than the blue curve of the baseline, which denotes that the window attention mechanism promotes weights in the network to be sensible on this identification task. After epoch 15, two curves from the baseline and DSDA are converged and become saturated.

In Figure 9, we also perform dataset sensitivity analysis according to the ratio of training dataset usage. Our proposed DSDA gives robustness until 60% usage of the training dataset by keeping the error rate below 30%, and then, the error rate increases. However, the baseline shows the robustness until 80% usage of the training dataset, which shows that our model learns more efficiently on the dataset and is less sensitive to dataset scarcity.

In Figure 10, to verify the practicality of the proposed model, we extended the HI task into a human localization task, where the MD signatures are extended to the longer size (i.e., 450 time stamps), and we perform time stamp wise human identifications. This can be thought of as human localization in MD signatures, which is more challenging and requires a fine-grained understanding of MD signatures. MD signatures are randomly selected in the validation set. The prediction of each stamp is performed on the center of 150 time stamp windows. DSDA traverses all the MD signatures and predicts every time stamp, which shows its predictions in the below Figure 10, where the y-axis denotes the probability of each target. Thus, five distributions corresponding to five targets are generated, and the above bar shows the ground-truth target for every signature. It is clearly confirmed that the distributions matched with the ground-truth target are highlighted for the given MD signatures. This denotes that DSDA is available to perform fine-grained human identification and is properly extended to the human localization task in radar signals.

## 5. Limitation

We would like to present two limitations from two different perspectives. The first limitation exists in terms of task. Our current experiments are mainly performed under a human identification task. However, as shown in Figure 10 of the paper, we found that this task can be extended up to the human detection task in the radar sequence by stiching MD signatures and localizing the human in them. Our further studies will include this extended analysis and possibilities of human detection. The second limitation is the lack of datasets for this study. We validate DSDA on the IDRad dataset. To build a general radar-based model, we should also validate more different radar datasets. In this respect, our further research will focus on how to establish model generality on radar signals.

## 6. Future Work

Our future works are three-fold. The first is that we will annotate the radar dataset to be suitable localization tasks. As shown in Figure 10, we further evaluate our DSDA in terms of human identification and localization tasks on the radar dataset, where DSDA shows enough applications on these tasks. The second is that we will modify the window. Currently, our window operates on the sliding window mechanism, but we will consider more diverse windowing methods. The third is that we will validate our DSDA on another radar dataset to be a general attention module for radar understanding.

## 7. Conclusions

We propose Dual-Scale Doppler Attention (DSDA) for human identification. DSDA adopts Temporal Window Attention (TWA) that attends fast-moving MD signatures of arms and legs and Holistic Window Attention (HWA) that attends slow-moving MD signatures of the torso. Our experimental results on the IDRad dataset show state-of-the-art performance and the qualitative results validate the efficiency of our proposed module. The model analysis including a sensitivity and confusion matrix shows DSDA’s robustness. Extended experiments incorporating radar-based human detection tasks show the flexibility of DSDA. From these experiments, our future works are more obvious and substantial. 

## Figures and Tables

**Figure 1 sensors-22-06363-f001:**
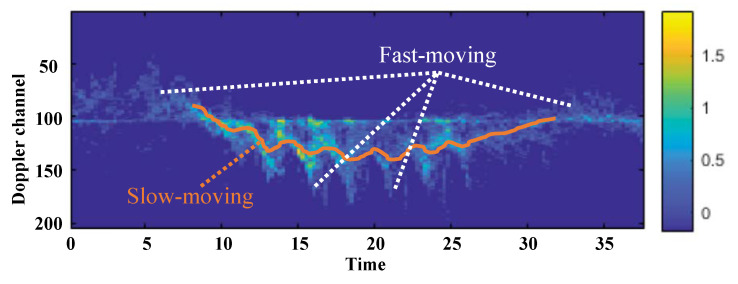
Micro-Doppler signature of human walking. When a person walks, the arms and legs generate fast-moving signatures represented by a white line, and the torso generates slow-moving signatures indicated by an orange line.

**Figure 2 sensors-22-06363-f002:**
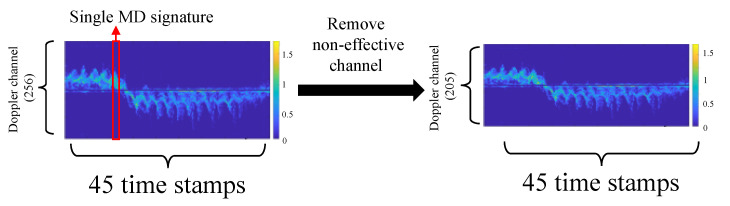
Illustration of Generating Micro-Doppler signature for Dual-Scale Doppler Attention.

**Figure 3 sensors-22-06363-f003:**
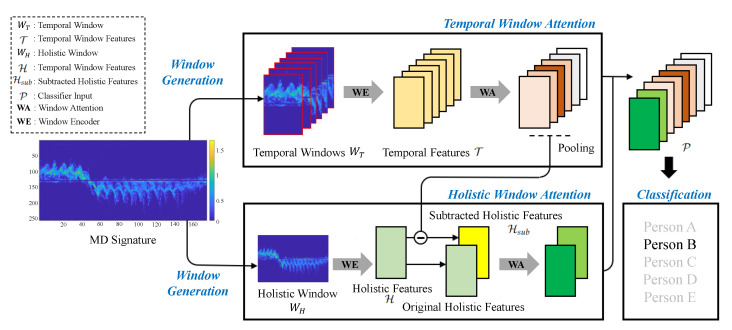
Illustration of Dual-Scale Doppler Attention (DSDA) for human identification. DSDA is composed of the following components: (1) Window Generation for designing temporal window and holistic window, (2) Temporal Window Attention for observing fast-moving features of arms and legs, (3) Holistic Window Attention for recognizing slow-moving features of the torso, and (4) Classification for classifying final targets.

**Figure 4 sensors-22-06363-f004:**
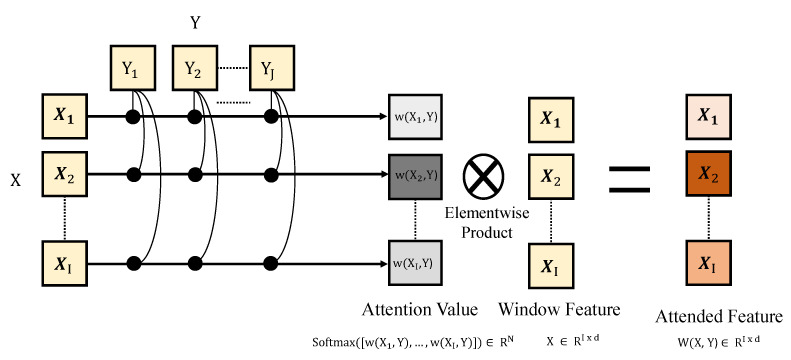
Illustration of Window Attention. The WA is performed on TWA and HWA according to the window features.

**Figure 5 sensors-22-06363-f005:**
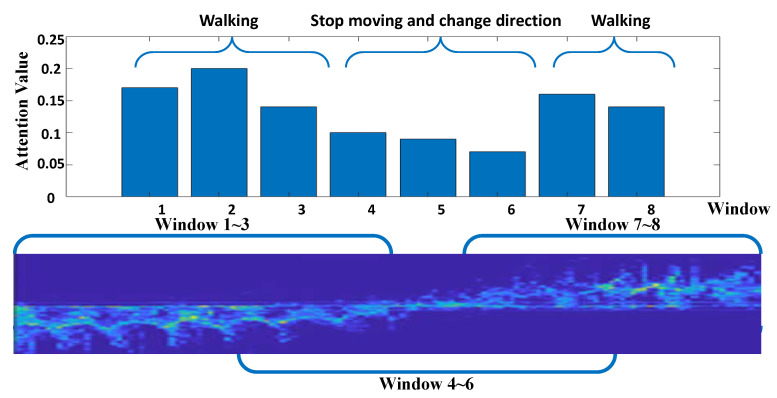
Attention weights according to temporal windows in TWA. The 8 windows are generated via traversing MD signatures with a sliding window composed of 45 time stamps and 15 strides.

**Figure 6 sensors-22-06363-f006:**
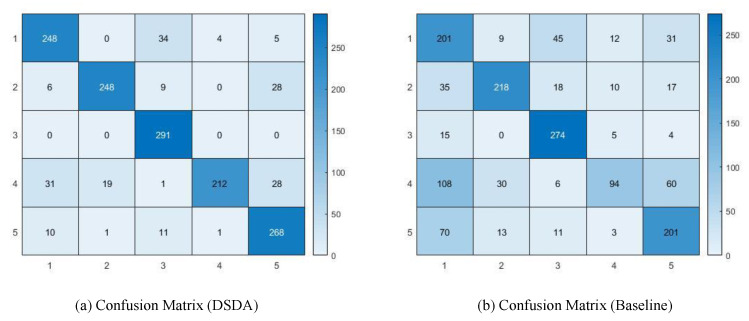
(**a**) Confusion matrix (5 × 5) of human identification from DSDA and (**b**) Confusion matrix (5 × 5) of human identification from baseline [10] on IDRad validation split. Vertical axis denotes ground-truth type and horizontal axis denotes predicted type.

**Figure 7 sensors-22-06363-f007:**
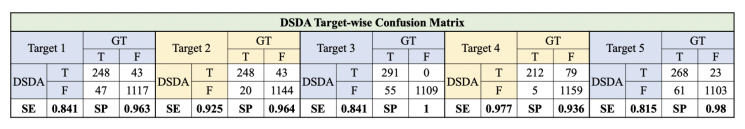
Target-wise Confusion Matrix to calculate sensitivity (SE) and specificity (SP) between DSDA and the ground-truth (GT) on the validation split of the IDRad Dataset (T: True, F: False).

**Figure 8 sensors-22-06363-f008:**
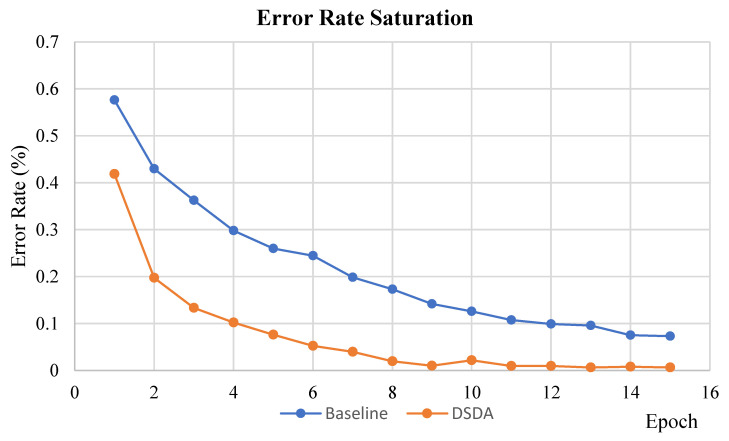
Error rate saturation along the epoch on the training dataset. The blue curve shows the saturation of baseline [10] and the orange curve shows the saturation of the proposed DSDA.

**Figure 9 sensors-22-06363-f009:**
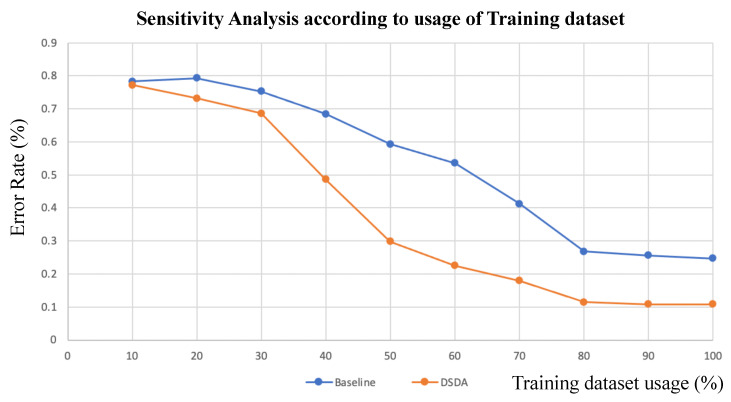
Sensitivity analysis of model performances according to the training dataset usage. The blue curve shows baseline [10], and the orange curve shows the proposed DSDA.

**Figure 10 sensors-22-06363-f010:**
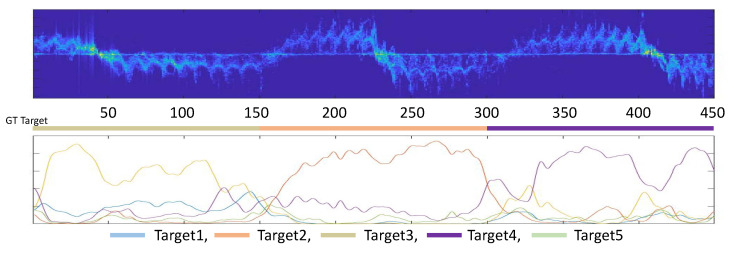
Extended experiment on MD signatures. The below distribution denotes time stamp wise target predictions, where this represents the availability of the human localization task of DSDA.

**Table 1 sensors-22-06363-t001:** Window Encoder Specifications for Temporal Window Attention.

Layer	Kernel Size	Stride	# of Filters	Data Shape
INPUT				(150×205×1)
Temporal Window				(3×50×205×1)
Conv 1	(3,3)	(1,1)	8	(3×50×205×8)
ELU 1				(3×50×205×8)
MAXPooL 1	(2,3)	(2,3)		(3×25×68×8)
Conv 2	(3,3)	(1,1)	16	(3×25×68×16)
ELU 2				(3×25×68×16)
MAXPooL 2	(2,3)	(2,3)		(3×12×22×16)
Conv 3	(3,3)	(1,1)	32	(3×12×22×32)
ELU 3				(3×12×22×32)
MAXPooL 3	(3,1)	(3,1)		(3×4×22×32)
Conv 4	(3,3)	(1,1)	64	(3×4×22×64)
ELU 4				(3×4×22×64)
MAXPooL 4	(1,5)	(1,5)		(3×4×4×64)
Pooling				(3×1024)

**Table 2 sensors-22-06363-t002:** Window Encoder Specifications for Holistic Window Attention.

Layer	Kernel Size	Stride	# of Filters	Data Shape
INPUT				(150×205×1)
Holistic Window				(150×205×1)
Conv 1	(3,3)	(1,1)	8	(150×205×8)
ELU 1				(150×205×8)
MAXPooL 1	(6,3)	(6,3)		(25×68×8)
Conv 2	(3,3)	(1,1)	16	(25×68×16)
ELU 2				(25×68×16)
MAXPooL 2	(2,3)	(2,3)		(12×22×16)
Conv 3	(3,3)	(1,1)	32	(12×22×32)
ELU 3				(12×22×32)
MAXPooL 3	(3,1)	(3,1)		(4×22×32)
Conv 4	(3,3)	(1,1)	64	(4×22×64)
ELU 4				(4×22×64)
MAXPooL 4	(1,5)	(1,5)		(4×4×64)
Pooling				(1×1024)

**Table 3 sensors-22-06363-t003:** Classifier detailed specifications.

Layer	Kernel SIZE	Stride	# of Filters	Data Shape
INPUT				((3+2)×1024)
Pooling	(3,3)	(1,1)	8	(5120)
Linear				(128)
ELU	(3,3)	(1,1)	16	
Dropout 4	(1,5)	(1,5)		
Linear				5

**Table 4 sensors-22-06363-t004:** Error rate (%) on the IDRad Dataset.

Methods	Validation	Test
Baseline [10]	24.70	21.54
Baseline [10] (reproduced on 150 time stamp)	22.42	20.37
PCA SVM	37.67	32.91
LSTM based model	22.83	18.42
DSDA	**10.87**	**8.65**

**Table 5 sensors-22-06363-t005:** Ablation study on model variants of DSDA on the validation split of IDRad.

Model Variants	Error Rate (%)
Full DSDA	10.87
w/o TWA	24.32
w/o HWA	23.94
w/ stride s=10, window L=25	12.99
w/ stride s=20, window L=25	13.71
w/ stride s=5, window L=15	14.73
w/ stride s=10, window L=15	14.80
w/ stride s=5, window L=35	11.93
w/ stride s=10, window L=35	13.45
Multi-Scale (M = 3)	10.87
Multi-Scale (M = 4)	12.79

**Table 6 sensors-22-06363-t006:** Comparison of attention methods in WA on the validation split of IDRad.

Attention Method for Calculating Similarity		Error Rate (%)
A∗B	$w\left(x_{i},Y\right)=\sum_{j=1}^{J}\left(\mathrm{ELU}\left(W_{1}x_{i}\right)\mathrm{ELU}{\left(W_{2}y_{j}\right)}^{T}\right)$	10.87
A+B	$w\left(x_{i},Y\right)=\sum_{j=1}^{J}\left(\left[\mathrm{ELU}\left(W_{1}x_{i}\right)+\mathrm{ELU}{\left(W_{2}y_{j}\right)}^{T}\right]W_{add}\right)$	11.28
A;B	$w\left(x_{i},Y\right)=\sum_{j=1}^{J}\left(\left[\mathrm{ELU}\left(W_{1}x_{i}\right);\mathrm{ELU}{\left(W_{2}y_{j}\right)}^{T}\right]W_{cat}\right)$	11.02

**Table 7 sensors-22-06363-t007:** Kappa Index Analysis of DSDA on the validation split of IDRad.

Human Identification		Baseline [10]		Total
		True	False	
DSDA	True	1082	246	1328
	False	40	122	162
Total		1122	368	1490

## Data Availability

Not applicable.

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
