# Peer review of "Dual-Scale Doppler Attention for Human Identification"

_sensors, 2022, doi:10.3390/s22176363_

Round 1
Reviewer 1 Report
Authors use micro doppler (MD) signatures for human identification using an attention network. Authors suggest that radar signals can be a good alternative to vision-based human identification systems. For this purpose, they use a small radar dataset called IDRad (only contains 5 individuals) for human re-identification. The proposed work consists of a dual-scale doppler attention network that captures human gait characteristics using temporal window attention (TWA) (for fast moving MS signatures) and holistic window attention (HWA) (for slow moving MD signatures). It is not clear how much novel the proposed approach is. Novelty needs to be clarified very clearly compared to other relevant works especially [16][17][30][32][35].
For a reader that is not very familiar with MD signatures, explanations are not clear. How MD signatures are created for each individual? Explain Fig 1 more clearly.
Related Work section needs more explanations about especially the related works [16][17][30][32][35]. What is the difference, advantage, disadvantage of the proposed work from them?
Method section is very brief and lacks many details. In Fig 2, TWA and HWA architectures are presented. However, from this figure, CNN architectures are not clear. Authors can create two separate figures more for TWA and HWA that shows convolution, pooling layers, dropout layers operations, input and outputs of the networks. How the attention is provided in the proposed TWA and HWA? Needs clarfication. Then in the subsequent text and equations, you can refer to these new figures. In its current form, it is difficult to understand the novelty of the attention networks.
Evaluations are conducted on a small dataset that consists of 5 individuals. Authors compare their work with [16] but it is important to compare other important and relevant works such as [30][32][35]? If their is a gain over these related works then we can judge its contribution.
How error rate is calculated? Give a formula.
Is it possible to compare the work with vision-based human re-identification? Is there a dataset for this? Does MD signals provide any advantage compared to camera based systems?
Ablation study is good and shows results for various combinations of the proposed work.
Conclusions need to be expanded. It is very short in its current form.
Author Response
We greatly appreciate Reviewer 1 for his/her very constructive and detailed comments. We address the questions of the Reviewers 1 by attached .pdf file to give detailed explanation with figure and table.

Reviewer 2 Report
In this manuscript, the authors propose a method based on deep learning and transformers to recognize individuals based on gait, captured by Doppler signals. The data is organized in the form of about 96,000 10-minute windows (training) and 25,000 5-minute windows (test). The signs are part of the IDRad database, where 5 people aged between 23 and 32 were filmed, with weights ranging from 60 to 90 kg and heights from 178 to 185 cm. These people were left free to walk around in a closed room. The trained deep model has 3 hidden CNN-like layers and 4 drop-out blocks in the linear output layer. The model was trained for 15 epochs, which was relatively little, but it is understood that the authors had time constraints, despite having used a good machine with a GPU. The results look good and reproducible. However, its expression in the form of errors is a bit strange: the most common is to express the results as hits. I therefore recommend that the authors include the accuracy, kappa index, sensitivity, specificity, and area under the ROC curve of the proposed model and state-of-the-art models for better comparison. I also suggest that authors modify their discussions accordingly with the introduction of these additional metrics. It is also important to discuss the limitations of the proposal, since a single validation experiment was carried out, when it would be better to have done at least 30 experiments, given that the initialization of deep networks, like any neural networks, is random. Otherwise, the work is well organized, the results are honest and the references are correct. The methodological aspects are clear enough to reproduce the proposal.
Author Response
We greatly appreciate Reviewer 2 for his/her very constructive and detailed comments. We address the questions of the Reviewers 2 by attached .pdf file to give detailed explanation with figure and table.

Reviewer 3 Report
The application of radar-based micro-doppler signature for human identification, similar to gait recognition, is a recent topic for identification. The proposed coarse-fine feature extraction strategy for deep learning is novel.
The architecture of the deep learning network has to be plotted, showing the number of layers for each functional blocks and the input and output dimensions at each layer. Otherwise the manuscript is hard to follow. The authors stated that the dimension of the hidden layer is 1024 in Section 4.2. This is ambuiguous.
What is the depth of the network? It is claimed to be deep convolutional network, and how deep is the network? If it is less than five layer, it is better called convolutional network. Classical deep neural networks usually has hundreds of layers.
Those math symbols need to be put in Figure 2 for easy understanding of the equations.
The format of the references is not consistent.
Author Response
We greatly appreciate Reviewer 3 for his/her very constructive and detailed comments. We address the questions of Reviewers 3 by attached .pdf file to give a detailed explanation with table.

Round 2
Reviewer 1 Report
Authors performed extensive revisions and addressed my comments.
Reviewer 3 Report
Accept